# Nucleus Accumbens-Associated Protein 1 Binds DNA Directly through the BEN Domain in a Sequence-Specific Manner

**DOI:** 10.3390/biomedicines8120608

**Published:** 2020-12-14

**Authors:** Naomi Nakayama, Gyosuke Sakashita, Takashi Nagata, Naohiro Kobayashi, Hisashi Yoshida, Sam-Yong Park, Yuko Nariai, Hiroaki Kato, Eiji Obayashi, Kentaro Nakayama, Satoru Kyo, Takeshi Urano

**Affiliations:** 1Department of Biochemistry, Shimane University School of Medicine, Izumo, Shimane 693-8501, Japan; n-nakayama@u-shimane.ac.jp (N.N.); gsakashi@med.shimane-u.ac.jp (G.S.); nariai@med.shimane-u.ac.jp (Y.N.); hkato@med.shimane-u.ac.jp (H.K.); eijioba@med.shimane-u.ac.jp (E.O.); 2Department of Health and Nutrition, The University of Shimane, Izumo, Shimane 693-8550, Japan; 3Institute of Advanced Energy, Kyoto University, Kyoto 606-8501, Japan; nagata.takashi.6w@kyoto-u.ac.jp; 4Graduate School of Energy Science, Kyoto University, Kyoto 606-8501, Japan; 5Institute for Protein Research, Osaka University, Suita, Osaka 565-0871, Japan; naohiro@protein.osaka-u.ac.jp; 6Protein Design Laboratory, Graduate School of Medical Life Science, Yokohama City University, Yokohama, Kanagawa 230-0045, Japan; h-yosida@yokohama-cu.ac.jp (H.Y.); park@yokohama-cu.ac.jp (S.-Y.P.); 7Department of Obstetrics and Gynecology, Shimane University School of Medicine, Izumo, Shimane 693-8501, Japan; kn88@med.shimane-u.ac.jp (K.N.); satoruky@med.shimane-u.ac.jp (S.K.)

**Keywords:** nucleus accumbens-associated protein 1 (NAC1), BEN (BANP, E5R and NAC1) domain, sequence-specific DNA-binding protein, solution NMR structure, isothermal titration calorimetry (ITC)

## Abstract

Nucleus accumbens-associated protein 1 (NAC1) is a nuclear protein that harbors an amino-terminal BTB domain and a carboxyl-terminal BEN domain. NAC1 appears to play significant and diverse functions in cancer and stem cell biology. Here we demonstrated that the BEN domain of NAC1 is a sequence-specific DNA-binding domain. We selected the palindromic 6 bp motif ACATGT as a target sequence by using a PCR-assisted random oligonucleotide selection approach. The interaction between NAC1 and target DNA was characterized by gel shift assays, pull-down assays, isothermal titration calorimetry (ITC), chromatin-immunoprecipitation assays, and NMR chemical shifts perturbation (CSP). The solution NMR structure revealed that the BEN domain of human NAC-1 is composed of five conserved α helices and two short β sheets, with an additional hitherto unknown N-terminal α helix. In particular, ITC clarified that there are two sequential events in the titration of the BEN domain of NAC1 into the target DNA. The ITC results were further supported by CSP data and structure analyses. Furthermore, live cell photobleaching analyses revealed that the BEN domain of NAC1 alone was unable to interact with chromatin/other proteins in cells.

## 1. Introduction

Nucleus accumbens-associated protein 1 (NAC1), encoded by the NACC1 gene, is a nuclear protein that encompasses an amino-terminal BTB (broad complex, tramtrack, bric-à-brac) and a carboxyl-terminal BEN (BANP, E5R and NAC1) domain. NAC1 was originally identified and cloned as a cocaine-inducible transcript from the nucleus accumbens, a unique forebrain structure involved in reward motivation and addictive behavior [1]. NACC1 was also identified as a cancer-associated BTB gene by serial analysis of gene expression in ovarian cancer cells [2]. NAC1 is significantly overexpressed in several types of carcinomas, including ovarian, colorectal, breast, renal cell, cervical, and pancreatic carcinomas, is associated with tumor growth and survival, and increases the resistance of tumor cells to chemotherapy [2,3,4,5,6,7,8,9,10,11,12,13,14]. Furthermore, NAC1 was shown to be important for the pluripotency of embryonic stem cells [15,16] through direct transcriptional regulation of c-Myc [17]. In addition, it was recently demonstrated that NAC1 promotes mesendodermal and represses neuroectodermal fate selection in embryonic stem cells, in cooperation with the pluripotency transcription factors, Oct4, Sox2, and Tcf3 [18,19] and that NAC1 is critical for efficient iPSC generation [20]. NACC1 knockout mouse embryos and newborns exhibit a lower survival rate compared to wild-type embryos and newborns, and surviving mice showing defective bony patterning in the vertebral axis [21]. NAC1 has also been reported to be an important component of RIG-I-like receptor mediated innate immune responses against viral infection [22].

The BTB domain is a ~120-amino-acid highly conserved motif that mediates homodimerization and/or heterodimerization and interacts with other proteins [23,24]. NAC1 homodimerizes through its BTB domain [25,26] and heterodimerizes with Myc-interacting zinc-finger protein 1 through the respective BTB domains [27,28,29]. Most BTB proteins contain other C-terminal functional domains, such as DNA-binding C2H2/Krüppel-type zinc fingers.

The BEN domain (BEND) in metazoans and some viruses typically comprises 90–100 amino acid residues. Secondary structure analysis using multiple alignment predicted four α-helices with conserved hydrophobic residues [30]. The human genome encodes nine BEN domain-containing proteins. These proteins can be divided into two distinct groups: one group contains a BTB domain (NAC1 and NAC2 (also known as BTBD14A and RBB)), whereas the other group lacks other recognizable protein domains (i.e., they are BEN-solo proteins [31]) such as BANP (also known as BEND1 and SMAR1) and BEND2, -3, -4, -5, -6, and -7. BEND2 and BEND3 harbor tandem copies of two and four BEN domains, respectively. Based on characterized proteins possessing a BEN domain, it has been predicted that the BEN domain is involved in chromatin organization and transcriptional regulation by mediating protein–DNA and protein–protein interactions [30,32,33]. NAC1 functions as a transcriptional repressor through its association with REST corepressor 1 (also known as CoREST) [32] and with histone deacetylases HDAC3 and HDAC4 [17,34]. NAC2 functions as a transcriptional repressor through its association with nucleosome remodeling and deacetylase complex containing HDAC1 and HDAC2 [35]. The activity of BANP as a transcriptional repressor and candidate tumor suppressor involves the BEN domain in molecular interactions with SIN3-histone deacetylase complex containing HDAC1 [36]. BEND3, a quadruple BEN domain-containing protein, is a key factor that associates with chromatin remodeling complexes and modulates gene expression and heterochromatin organization [33,37,38]. BEND5, a neural BEN-solo protein, functions as a sequence-specific transcription repressor that regulates neurogenesis [39]. BEND6, a neural BEN-solo protein, acts as a corepressor of notch transcription factor [40].

## 2. Experimental Section

### 2.1. Cloning

For glutathione S-transferase (GST) fusion, NAC1 (2–527)/pMXs-FHG and NAC1 (2–250)/pMXs-FHG (41) were digested with BamHI and EcoRI. The fragments of human NAC1 cDNA encoding residues 2–527 (full-length, removal of the first methionine) and 2–250 were subcloned into pGEX-4T-2 (GE Healthcare, Piscataway, NJ, USA). The C-terminal fragment of human NAC1 cDNA encoding residues 251-527 was cloned into pGEX-4T-2 by polymerase chain reaction (PCR) using NAC1 (2–527)/pMXs-FHG as template. The NAC1(L432N) mutant was created by standard double PCR mutagenesis and subcloned into pGEX-4T-2. Human NPM1 full-length cDNA (CCDS4376.1) obtained by reverse transcribed PCR using the total RNA from HeLa cells was cloned into pGEX-4T-2.

To generate hexahistidine (His) tagged protein, NAC1 (2–527)/pGEX-4T-2 was digested with BamHI and EcoRI, then subcloned into pET28HisTEV [41]. The C-terminal fragment of human NAC1 cDNA encoding residues 322-485 was cloned into pET28HisTEV by PCR using NAC1 (2–527)/pMXs-FHG as template. All PCR-amplified cDNA products were fully sequenced using a 3130 genetic analyzer (Applied Biosystems/ThermoFisher Scientific, Waltham, MA, USA) to confirm their sequence and to verify the absence of secondary point mutations.

### 2.2. Screening of DNA-Binding Sequences

The random oligonucleotide pool N26 (5′-CAGGTCAGTTCAGCGGATCCTGTCG(N)26GAGGCGAATTCAGTGCAACTGCAGC-3′) contains a synthetic single-stranded random 26-base sequence flanked by two PCR primer sequences (N26-1S, 5′-CAGGTCAGTTCAGCGGATCCTGTCG-3′; N26-2AS, 5′-GCTGCAGTTGCACTGAATTCGCCTC-3′). The flanking sequences contain BamHI and EcoRI restriction sites to facilitate subsequent cloning. The random oligonucleotide pool (1.2 μg) was precleared with recombinant GST-NAC1(L432N), then incubated for 30 min at room temperature with GST-NAC1 in 500 μL of binding buffer (20 mM HEPES/KOH pH 7.9, 50 mM KCl, 2 mM MgCl2, 0.5 mM EDTA, 10% glycerol, 1 mg/mL bovine serum albumin, 2 mM dithiothreitol, and 10 mg/mL poly(dI/dC)). After washing three times with the binding buffer, samples were resuspended in 100 μL distilled water and boiled for 10 min to elute bound DNA. Following centrifugation, 10 μL of the supernatant was used as template for PCR with the forward primer N26-1S and reverse primer N26-2AS using the following PCR conditions: 20 cycles of 1 min at 94 °C, 1 min at 62 °C, and 1 min at 72 °C. The PCR products were then purified using a PCR purification kit (Qiagen, Hilden, Germany), and resuspended in distilled water. Half the volume of each purified PCR product was subsequently used for the next cycle of selection. Four sequential repeats of selection were followed by PCR and the final selected DNA fragments were digested by BamHI and EcoRI, subcloned into pBluescript SK II (Stratagene, La Jolla, CA, USA), and the sequences were determined using T7 primer. The sequence data are shown aligned using the same orientation (as primed from the BamHI side) (Figure 1A).

### 2.3. Bioinformatics

Amino acid sequence alignment of the BEN domain from human BEN-domain containing proteins were conducted using Clustal Omega (http://www.ebi.ac.uk/Tools/msa/clustalo/).

PCR-assisted random oligonucleotide selection identified 18 independent clones that bound to GST-NAC1 and these sequences were entered into the MEME program (http://meme-suite.org/tools/meme) (Figure 1C). The relevant parameters were set such that each sequence was used once to generate a motif between 6–26 nucleotides long.

### 2.4. Oligonucleotide Pull-Down Assay

Sense oligonucleotides for the GADD45GIP1 promoter, with biotin added to their 5′-end, were synthesized by FASMAC (Kanagawa, Japan). The sequences for the oligonucleotides were as follows: Biotin-GP1, biotin-5′-TGTGTGTGTATGCATGTATGTATTTATT-3′ (wild type, sense) and 5′-AATAAATACATACATGCATACACACACA-3′ (wild type, antisense); Biotin-GP1 5′-mut, biotin-5′-TGTGTGTGTCATCATGTATGTATTTATT-3′ (sense) and 5′-AATAAATACATACATGATGACACACACA-3′ (antisense); Biotin-GP1 mut, biotin-5′-TGTGTGTGTATGGCATTATGTATTTATT-3′ (sense) and 5′-AATAAATACATAATGCCATACACACACA-3′ (antisense); Biotin-GP1 3′-mut, biotin-5′-TGTGTGTGTATGCATGACGGTATTTATT-3′ (sense) and 5′-AATAAATACCGTCATGCATACACACACA-3′ (antisense). Each pair of oligonucleotides was annealed following standard protocols. Bacterially expressed and purified GST-NAC1 (5 μg) were precleared using streptavidin-magnetic beads (20 μL/sample, Dynabeads M-280, ThermoFisher Scientific) for 1 h at 4 °C. The precleared supernatant was incubated with 100 pmol of biotinylated double-stranded DNA (dsDNA) oligonucleotides and 10 μg of poly(dI/dC) for 30 min at 4 °C. DNA-bound proteins were collected using 30 μL of streptavidin-magnetic beads for 1 h at 4 °C. After washing three times, the bound proteins were eluted from the beads in SDS-PAGE sample buffer and resolved by SDS-PAGE, followed by Coomassie blue staining of the gel.

### 2.5. Antibody and Chromatin Immunoprecipitation (ChIP) Assay

To generate rabbit polyclonal antibodies, the region encompassing amino acids 251-527 of human NAC1 was expressed as His-tagged proteins in Escherichia coli. The anti-NAC1 antibodies were affinity-purified on protein coupled to CNBr-activated Sepharose 4B (GE Healthcare, Buckinghamshire, UK).

Human ovarian OV207 cells were cross-linked with a final concentration of 1% (*v*/*v*) formaldehyde for 10 min at 37 °C, then the cells were incubated for 5 min at 37 °C in 0.125 M glycine to stop the cross-linking reaction. After washing twice with ice-cold PBS, the cells were lysed with 200 μL of SDS-lysis buffer (1% SDS, 10mM EDTA, 50mM Tris-HCl, pH 8.1) and protease inhibitors for 10 min on ice, then 1800 μL of ChIP buffer (0.01% SDS, 1.1% Triton X-100, 1.2mM EDTA, 16.7mM Tris-HCl, pH 8.1, 167mM NaCl) was added. A Branson sonifier 250 was used to shear the genomic DNA by sonification. After removal of cellular debris by centrifugation and digestion of the RNA with RNase A, equal amounts of DNA were incubated with 3 μg of anti-NAC1 antibody or control IgG antibodies previously bound to anti-rabbit IgG-coupled magnetic beads (Dynabeads M-280, ThermoFisher Scientific). After extensive washing, the precipitated DNA fragments were eluted. Precipitated DNA was analyzed by PCR using the following primers: PSAT1, 5′-GGCAGGTGGTCAACTTTGG-3′ (forward) and 5′-GAACACTAATGCCAACTCC-3′ (reverse); AZU1, 5′-TTTCCATCAGCAGCATGAGC-3′ (forward) and 5′-TATCGTCACGCTGCTGGTG-3′ (reverse); β-actin, 5′-CCAACCGCGAGAAGATGACCC-3′ (forward) and 5′-CGTCACCGGAGTCCATCACGA-3′ (reverse). Each PCR product was quantified using a Takara Thermal Cycler Dice RealTime System (Kusatsu, Shiga, Japan). ChIP experiments were performed at least in triplicate on independent biological samples. The value for each sample, as shown in Figure 2C, was normalized using the value for β-actin.

### 2.6. Isothermal Titration Calorimetry (ITC) Experiments

All calorimetric titrations were carried out on iTC200 calorimeters (MicroCal, Malvern Biosciences, Columbia, MD, USA). Protein samples were dialyzed against the buffer containing 20 mM HEPES (pH 7.0) and 0.1M NaCl with 1mM Tris (2-carboxyethyl)phosphine. The sequences for the oligonucleotides were as follows: GP1 [14], 5′-GTATGCATGTATGT-3′ (wild type, sense) and 5′-ACATACATGCATAC-3′ (wild type, antisense), GP1 mut, 5′-GTATGGCATTATGT-3′ and 5′-ACATAATGCCATAC-3′, de novo motif 1, 5′-AACCGCCGCCAA-3′ and 5′-TTGGCGGCGGTT-3′, de novo motif 2, 5′- AACCCCGCCCCAA-3′ and 5′- TTGGGGCGGGGTT-3′. The sample cell was filled with a 50 μM solution of dsDNA or ssDNA of GP1, dsDNA of GP1 mut, de novo motif 1 and motif 2, and the injection syringe was filled with 500 μM titrating NAC-1322-485. For iTC200, each titration typically consisted of a preliminary 0.4-μL injection followed by 19 subsequent 2-μL injections every 150 s. All of the experiments were performed at 20 °C. Data for the preliminary injection, which were affected by diffusion of the solution from and into the injection syringe during the initial equilibration period, were discarded. Binding isotherms were generated by plotting heats of reaction normalized by the moles of injected protein versus the ratio of the total injected one to total DNA per injection. The data were fitted using Origin software (Northampton, MA, USA).

### 2.7. NMR Spectroscopy

Escherichia coli strain BL21 (DE3) RIPL CodonPlus was transformed with NAC1 (322–485)/pET28HisTEV. Cells were induced with 0.5 mM IPTG, then grown overnight at 15 °C in M9 minimal medium containing 1 g/L 15N-NH4Cl (ISOTEC, Canton, GA, USA) and 5 g/L 13C-glucose (ISOTEC) as nitrogen and carbon sources, respectively. Harvested cells were resuspended in Ni-NTA binding buffer (20 mM Tris-HCl, pH 8.0, 1M NaCl, 25 mM imidazole, 10 mM β-mercaptoethanol) and lysed using an EmulsiFlex homogeniser (Avestin, Ottawa, ON, Canada). After centrifugation, the supernatant was loaded onto Ni-NTA agarose (Qiagen, Hilden, Germany) columns equilibrated with the same buffer. Protein was eluted using a 25–500 mM linear gradient of imidazole. Peak fractions were incubated overnight with His-tagged TEV protease at room temperature while dialyzing against Ni-NTA low-salt buffer (20mM Tris-HCl, pH 8.0, 150 mM NaCl, 25 mM imidazole, 10 mM β-mercaptoethanol). After complete cleavage, the sample was loaded onto a Ni-NTA agarose column to remove the His-tag, His-tagged TEV protease, and minor protein contaminants.

13C, 15N-labeled NAC1322-485 was concentrated to 500 μM and dissolved in 20 mM sodium phosphate buffer (pH 6.0) prepared using 95% 1H2O/5% 2H2O and containing 50 mM NaCl and 1 mM 1,4-DL-dithiothreitol (DTT). All NMR data were collected at 298 K on a Bruker AVANCE 600 MHz NMR spectrometer (Billerica, MA, USA) equipped with a cryogenic probe. NMR spectra were processed with NMRPipe/NMRDraw program [42]. Spectral analysis was performed with KUJIRA 0.984 [43], a program suite for interactive NMR analysis that works in conjunction with NMRView [44], and SPARKY 3, according to the methods described previously [45]. The backbone 1H, 15N, and 13C resonances of NAC1322-485 were assigned by standard double- and triple-resonance NMR experiments [46,47]: 2D 1H–15N heteronuclear single quantum correlation (HSQC), and 3D HNCO, 3D HN(CA)CO, 3D CBCA(CO)NH, and 3D HNCACB spectra.

Assignments of side chain resonances for nonaromatic residues were obtained by 2D 1H-13C HSQC and 3D HBHA(CO)NH, HC(CCO)NH, C(CCO)NH, H(C)CH correlation spectroscopy (COSY), H(C)CH-total correlated spectroscopy (TOCSY), and (H)CCH-TOCSY, whereas assignments for aromatic residues were performed using H(C)CH COSY, aided by 13C-edited nuclear Overhauser effect spectroscopy (NOESY)-HSQC spectra obtained using a mixing time of 80 ms. Distance restraints were derived from 3D 15N-edited and 13C-edited NOESY-HSQC spectra, and each was measured using a mixing time of 80 ms. Protein backbone φ, ψ and side chain χ1, χ2 torsion angle restraints were determined by chemical shift database analysis using the program TALOS+ [48] and by inspecting the pattern of intraresidual NOE intensities [49], respectively. The chemical shift data were deposited in BioMagResDB (BMRB ID: 36342).

An NMR titration experiment on NAC1322-485 with dsDNA was performed by recording 2D 1H–15N HSQC spectra at 298 K. The sequences for the 14-mer oligonucleotides derived from GP1 were 5′-GTATGCATGTATGT-3′ (sense) 5′-ACATACATGCATAC-3′ (antisense) and the oligonucleotide pair was annealed following standard protocols. Increasing amounts of unlabeled dsDNA were added to 75 μM 15N-labeled NAC1322-485 to obtain molar ratios of 1:0, 1:0.2, 1:0.4, 1:0.6, 1:0.8, 1:1, 1:1.2, and 1:1.5. The weighted chemical shift perturbations for backbone 15N and 1HN resonances at a ratio of [protein]:[dsDNA] = 1:1.5 were calculated as: Δδ = [(ΔδHN)2 + (ΔδN/6.5)2]1/2 [50].

### 2.8. Structure Calculation

Structure calculations for NAC1322-485 were performed using CYANA 2.1 [51,52,53]. The standard CYANA simulated annealing schedule was used with 40,000 torsion angle dynamics steps per conformer with 200 initial randomized conformers. The 40 conformers exhibiting the lowest final CYANA target function values were further energy minimized with AMBER 12 [54] using the AMBER 2003 force field and a generalized Born model, as described previously [45]. The force constants for distance, torsion angle, and ω angle restraints were set to 32 kcal mol-1 Å-2, 60 kcal mol-1 rad-1, and 50 kcal mol-1 rad-2, respectively. The 20 conformers that were most consistent with the experimental restraints were then used for further analyses. The final structures were validated and visualized by using the Ramachandran plot web server [55] and CHIMERA software [56,57]. Detailed experimental data and structural statistics are summarized in Appendix A. The final ensembles of 20 conformers were deposited in the Protein Data Bank (PDB ID: 7BV9).

### 2.9. Cell Culture and siRNA Transfection

The human ovarian cell line OV207 was a kind gift from Dr. Ie-Ming Shih (Johns Hopkins Medical Institutions, Baltimore, MD, USA). Cells were grown in Dulbecco’s modified Eagle’s medium (DMEM; Nissui Pharmaceutical, Tokyo, Japan) supplemented with 10% fetal bovine serum (Sigma, St. Louis, MO, USA). Stealth small interfering RNA (siRNA) against NAC1 (5′-CCGGCUGAACUUAUCAACCAGAUUG-3′) and control siRNA (5′-CACAUGAAGCAGCACGACUUCUUCA-3′) were purchased from ThermoFisher Scientific [12]. Cells were transfected with siRNA using Oligofectamine (ThermoFisher Scientific), according to the manufacturer’s instructions.

### 2.10. FRAP and FLIP Analyses

Fluorescent images were acquired using an FV1000 laser scanning confocal unit coupled to an inverted microscope (model IX81; Olympus, Tokyo, Japan) equipped with a ×100 oil immersion objective (UPLSAPO 100XO, NA 1.40, Olympus) and analysed using Fluoview software (Olympus). Cells were maintained at 37 °C in a Tokai Hit incubation system for microscopes (Tokai Hit, Shizuoka, Japan). Cells were analyzed on 35 mm glass-bottom dishes (IWAKI, Tokyo, Japan) in CO2-independent medium (ThermoFisher Scientific) to avoid medium acidification in the CO2-free atmosphere. A 473 nm laser (laser power: 0.1%) was used to excite GFP. A square region of interest (ROI) of 150 × 150 pixels (1 pixel = 0.124 µm) was scanned with a 2-line Kalman filter to obtain a noise-free image ideal for intensity measurements.

For FRAP experiments, five and 95 images (scanning time: 456 ms/frame) were acquired before and after bleaching, respectively. Photobleaching was performed by creating a bleach spot of 8 × 8 pixels using the Tornado mode of the FV1000 confocal unit (bleaching time: 250 ms; laser power: 100%). Using this bleaching condition, the bleaching constant (K) and the half width of the beam (w) were estimated to be 3.56 ± 0.432 and 1.63 ± 0.145, respectively. The intensities of the bleached area (B), unbleached area (U), and the background (bg) in each post-bleach image were measured to obtain the intensity ratio (Rn): Rn = (B − bg)/(U − bg). Rn values were divided by the average intensity ratio (R0) of the five pre-bleach images to obtain the normalized FRAP values (R): R = Rn/R0. We defined the time point when R reached 0.5 as t1/2. For each condition, the mean and SD of at least 35 individual FRAP values were calculated to draw the plots. Welch’s two sample t test was used in the FRAP analysis to calculate the *p*-values; ***, *p* < 0.001.

FLIP experiments were conducted using the time controller function of the FV1000, according to the manufacturer’s instructions. Five pre-bleach images were acquired before beginning 50 bleach-scan cycles. In a bleach-scan cycle, half of the ROI (150 × 75 pixels) was bleached once (laser power: 100%) before scanning a post-bleach image (150 × 150 pixels). Intensity ratios of the pre- and post-bleach images were calculated to obtain normalized FLIP values, as in the FRAP experiments. For each condition, the mean and SD values of at least 8 individual FLIP values were calculated to draw the plots.

### 2.11. Accession Numbers

Detailed experimental data and structural statistics have been deposited with the Protein Data bank under accession number 7BV9. The chemical shift data have been deposited in BioMagResDB (BMRB ID: 36342).

## 3. Results

### 3.1. NAC1 Binds DNA through Its BEN Domain

We have recently shown that NAC1 harbors an unusual bipartite-type nuclear localization signal to endow transcriptional regulator functions (Figure 1A) [58], and used live cell photobleaching analysis to show that a substantial fraction of NAC1 in the nucleus is associated with or interacts with nuclear proteins or chromatin [59]. However, whether NAC1 directly binds DNA, and if so, which domain of NAC1 is involved in binding, remains unknown. We performed gel mobility shift assays to investigate whether NAC1 protein directly binds DNA, using nucleophosmin (NPM1) as a positive control [60]. Linearized pBluescript SK II plasmid DNA was incubated with bacterially expressed and purified glutathione S-transferase (GST)-NAC1 protein. We found that the mobility of SKII DNA was slower than that of free plasmid DNA, whereas there was no difference in the mobility of SK II DNA following incubation with proteins such as GST alone or BSA, as shown in Appendix A. During the experiments, we noticed that bacterially expressed and purified NAC1 proteins are contaminated with the bacterial genome (Appendix A, lane 3). The size distribution of the bacterial genome was likely the result of physical shearing by sonication during cell lysis. For subsequent experiments we therefore used a simple and convenient assay without linearized plasmid DNA as a screening tool and denoted this assay ‘bacterial genome carry-over assay’.

To determine which domain of NAC1 is responsible for its DNA-binding properties, we divided NAC1 protein into two parts: the N-terminal half containing the BTB domain and nuclear localization signal (NLS), and the C-terminal half harboring the BEN domain (Figure 1A). Bacterial genome carry-over assays clearly showed that the C-terminal, but not the N-terminal half of NAC1 is contaminated with the bacterial genome (Appendix A). The BEN domain was identified by computational analysis and may mediate protein–DNA and/or protein–protein interactions during chromatin organization and transcription [30]. We hypothesized that the BEN domain directly binds DNA. We therefore attempted to generate full-length NAC1 protein mutants with amino acid substitutions of highly conserved residues in the BEN domain (Figure 1A). Full-length NAC1 carrying the L432N substitution was expressed in E. coli, purified, and then subjected to bacterial genome carry-over analysis. As shown in Appendix A, NAC1 (L432N) showed severely impaired carry-over activity. DNase or RNase treatment of the carry-over materials revealed that NAC1 exhibited a clear preference for DNA over RNA (Appendix A). These findings suggest that NAC1 directly binds DNA via the BEN domain.

### 3.2. Identification of the Consensus DNA-Binding Sequence of NAC1

We used a PCR-assisted random oligonucleotide selection approach to identify the consensus DNA-binding sequence for human NAC1 [61] (Figure 1B). Bacterially expressed GST-NAC1 protein on beads was washed with an excess volume of washing buffer containing 500 mM NaCl to prevent contamination by the bacterial genome. The purified protein was used to select preferred binding sequences from a random pool of dsDNA. Following precipitation of the NAC1-DNA complex using glutathione beads, we washed away unbound oligonucleotides, eluted the NAC1-DNA complex, and then PCR amplified the selected fragments through four rounds of selection. The background of oligonucleotides precipitated due to non-specific binding was reduced by preclearing the oligonucleotide pool with GST-NAC1 (L432N) beads in each round. We then cloned and sequenced all 51 oligonucleotide fragments (18 different clones) (Figure 1C, lower panel). All 18 oligonucleotides were used to generate a binding motif using MEME software [62]. Our results show that NAC1 selects the palindromic 6-bp motif ACATGT containing the core sequence CATG (Figure 1C, upper panel).

GST is known to form stable dimers. To eliminate the influence of GST-fusion, we checked the binding of NAC1 (251-527) without GST to the most frequently cloned sequence #8-5 using band shift experiments. As shown in Figure 1D, the dsDNA oligonucleotide derived from #8-5 was avidly bound by NAC1 without GST, clarifying that the binding of NAC1 to DNA is direct.

NAC1 has been reported to repress the transcriptional activity of human GADD45GIP1 [3,6]. In our hands, siRNA-mediated knockdown of endogenous NAC-1 resulted in increased levels of GADD45GIP1 protein [41]. The promoter region of human GADD45GIP1 was predicted by searching the Ensembl regulatory build database (ENSR00000343712) [63]. Visual examination of the promoter region upstream of ATG in human GADD45GIP1 revealed only one motif-like sequence within the region (Appendix A). We investigated whether NAC1 directly binds to the site detected on the GADD45GIP1 promoter using the NAC1 consensus in the context of its surrounding sequence by performing oligonucleotide pull-down assays. We incubated bacterially expressed and purified GST-NAC1 with biotin-labeled dsDNA oligonucleotides for wild type and mutant probes against GADD45GIP1 promoter, then performed pull-down assays and Coomassie blue staining to detect NAC1 in the DNA-protein complexes pulled down by the oligonucleotides. As shown in Figure 2A, we detected binding of NAC1 to wild type dsDNA oligonucleotide GP1. The core sequence mutation of CATG to GCAT in GP1 (GP1 mut) decreased the binding of the probe to NAC1, whereas mutations outside the core sequence, such as ATG to CAT (GP1 5′-mut) or TAT to ACG (GP1 3′-mut) in GP1, retained binding activity. These results indicate that the palindromic core sequence CATG is necessary to affect binding, whereas mutations outside the core sequence are not sufficient to decrease binding activity.

Next, we performed isothermal titration calorimetry (ITC) experiments to measure the binding affinity of NAC1322-485 to dsDNA oligonucleotide GP1. The titration curve clearly indicated the presence of two sequential and non-equivalent binding events (Figure 2B and Appendix A). The thermodynamic data of the first binding event showed that the positive binding enthalpy (ΔH1 = 1.8 kcal/mol) is compensated by entropy contribution (ΔS1 = 37.1 cal/mol/deg), which corresponds to the binding affinity (Kd1, 0.16 μM). This suggests that the interaction between protein and DNA is formed by hydrophobic contacts through base unspecific manner. The ITC data of the second event showed the negative binding enthalpy (ΔH1 = −30.4 kcal/mol) and entropy contribution (ΔS1 = −80.2 kcal/mol/deg), which corresponds to the binding affinity (Kd2, 18.0 μM), indicating the sequence specific base recognition involving in structural conformational change of protein. Titration of the ssDNA into the protein (Figure 2B, right panel) elucidated a double exothermic binding event as well as the case of that using GP1 dsDNA, but in contrast, both are similar binding manner, corresponding to that of Kd1 for GP1 dsDNA, with binding affinity (Kd1, 0.2 μM, Kd2, 0.46 μM). These results clarified the binding preference of NAC1 to dsDNA over ssDNA. Also, ITC experiment was performed using GP1 (mut) dsDNA. Although binding of GP1 (mut) dsDNA to NAC1 was weaker than that of GP1 shown by pull down assay (Figure 2A, lane 4), ITC data elucidated that GP1 (mut) dsDNA binds to NAC1 in similar binding manner showing two phase interactions. Binding affinity of first event is 135 μM, ∼840-fold larger binding affinity than that of Kd1 for GP1, whether that of second one is 0.37 μM, ∼48-fold smaller binding affinity than that of Kd2 for GP1. These results indicate that NAC1 binds DNA, especially recognizing a palindromic core sequence CATG involved in protein structural change. Furthermore, binding experiments were measured using two different motifs dsDNA reported by Malleshaiah et al. as NAC1 binding motifs elucidated by ChIP methods [18]. Both binding curves show interactions through base unspecific manner corresponding to that of Kd1 for GP1 (Kd for de novo motif 3.0 μM, Kd for de novo motif 2, 0.82 μM) (Appendix A). Therefore, the preferable sequence for NAC1 is GP1 more than de novo motif 1 and motif 2.

We conducted a chromatin immunoprecipitation (ChIP) assay to verify whether NAC1 binds to promoter regions in cells to regulate gene expression using the NAC1 consensus sequence. We were unable to generate suitable primers for the ChIP assay against GADD45GIP1 promoter because of composition, complexity and interspersed repeats content. The promoter regions of the PSAT1 and AZU1 genes were therefore chosen for further analyses for several reasons. First, one of the top ten could be selected from genes by microarray experiments upon NAC1 knockdown in the human ovarian OV207 cell line compared with control siRNA treated OV207 cells (data not shown). Second, promoter regions (PSAT1, ENSR00001306399; AZU1, ENSR00000341277, Appendix A) predicted by the Ensembl regulatory build database harbored the consensus DNA-binding sequence of NAC1. Third, we could construct appropriate primers for the ChIP assay against these promoters. Fourth, as detected by real-time quantitative PCR (RT-qPCR), the expression of PSAT1 mRNA was significantly reduced upon NAC1 knockdown in OV207 cells. However, the expression of AZU1 mRNA was up-regulated (Figure 2C, left panel). ChIP assay using NAC1 antibody detected the binding of NAC1 to the promoter regions of PSAT1 and AZU1 (Figure 2C, right panel). These results confirm the in vitro and in cell validity of the NAC1-binding consensus sequence determined by our selection procedure.

### 3.3. Solution NMR Structure of the BEN Domain of NAC1

The solution structure of NAC1322-485, a construct encompassing the BEN domain, was determined by multidimensional heteronuclear NMR [47]. Nearly complete resonance assignments were obtained for this construct using standard double and triple resonance experiments. NMR structures were generated using a combination of CYANA distance geometry calculations [51,52,53] and restrained molecular dynamics refinements in AMBER [54]. The 20 conformers with the lowest restraint violation energy were selected for the final representative ensemble and are shown in Figure 3A and Appendix A. The BEN domain of human NAC1 was determined to be composed of five conserved α helices and two short β sheets, with one additional N-terminal α helix (hereafter abbreviated α0).

NMR chemical shifts are very sensitive to changes in protein structure, and their variation upon titration of a protein with its binding partner allows accurate mapping of the binding site [47]. A dsDNA 14-mer GP1S was gradually titrated into a 75 μM solution of 15N-labeled NAC1322-485 to a final ratio of 1.5:1 of GP1S to NAC1. Binding was monitored by collecting the 1H–15N HSQC spectra of NAC1 at each titration point. An overlay of eight spectra, one for each titration point in the series, is shown in Figure 3B, where the free NAC1 spectrum is shown in grey, and the NAC1 spectrum in the presence of a 1.5× molar excess of dsDNA is shown in red. The five insets depict five selected spectral regions and clearly show the disappearance of peaks or large chemical shift changes (color gradient from grey to red) upon the addition of GP1S dsDNA oligonucleotide.

The weighted CSPs of the backbone amide nitrogens and protons between the free and bound forms of NAC1 were plotted versus the residue number (Figure 3C). Most residues whose peaks disappeared or showed a large change in frequency (Δδ > 0.13 ppm) upon the addition of GP1S dsDNA oligonucleotide were in the BEN domain, and particularly in the region connecting α3 and α4 (residues 413–434). It is noteworthy that the residues in this region except Leu-432 and Asp-433, are not conserved between human BEN domain-containing proteins, whereas the regions of human BEN domain-containing proteins harbor one to seven positively charged residues (Arg-421, Arg-428, Arg-429, and Lys-431 in human NAC1) (Figure 1A). As seen in Figure 3B,C, only a subset of peaks changed position upon the addition of dsDNA, indicating a specific interaction between the BEN domain of NAC1 and the GP1S dsDNA oligonucleotide.

### 3.4. The BEN Domain of NAC1 Alone Cannot Interact with Chromatin/Other Proteins in Living Cells

To study the dynamics and function of the BEN domain of NAC1 in living cells, we generated HeLa cell lines stably expressing GFP-NAC1 (251-527) or GFP-NAC1 (251-527, L432N) (Appendix A). NAC1 (L432N), in which a conserved leucine residue within the BEN domain of NAC1 is replaced with asparagine, severely impaired DNA-binding activity (Appendix A). Asynchronized HeLa cells stably expressing GFP-NAC1 (251-527) or GFP-NAC1 (251-527, L432N) were subjected to fluorescence recovery after photobleaching (FRAP) analysis (Figure 4A). In parallel, HeLa cells stably expressing a freely diffusing GFP, GFP-NAC1, or a very stable chromatin binding histone H3 fused with GFP (GFP-histone H3), were analyzed for comparison [59]. NAC1 (251-527) showed significantly faster recovery kinetics (t½ = 1.06 ± 0.52 s) compared to full-length NAC1 (t½ = 3.19 ± 1.21 s) (Figure 4B,C). In contrast, there was no difference in the nuclear mobility and kinetic properties of NAC1 (251-527) and NAC1 (251-527, L432N).

We next employed a complementary approach, fluorescence loss in photobleaching (FLIP) (Figure 4D). Repetitive bleaching of the GFP-NAC1 signal in one half of the nucleus led to gradual fluorescence loss in the other half of the nucleus, and resulted in a complete loss of fluorescence from the entire nucleus within 3 min (Figure 4E, GFP-NAC1, compare GFP and GFP-histone H3). Consistent with the FRAP data, this result indicates that NAC1 in the nucleus is highly mobile and rapidly exchanges with unbleached GFP-NAC1 molecules in the nucleus, although its diffusion is slower than that of freely diffusing GFP. GFP-NAC1 (251-527) showed a considerably faster loss of fluorescence in the unbleached half of the nucleus compared to wild-type GFP-NAC1, and there was no difference between NAC1 (251-527) and NAC1 (251-527, L432N) (Figure 4E). Taken together, these findings suggest that the BEN domain of NAC1 alone is likely to not interact with chromatin/other proteins as avidly as full-length protein.

## 4. Discussion

On the basis of the contextual conservation of BEN domain-containing proteins, the BEN domain has been predicted to mediate chromatin organization and transcriptional regulation by mediating protein–DNA and protein–protein interactions [30]. The Drosophila genome encodes three BEN domain-containing proteins: Insensitive (Insv), Bsg25A (Elba1), and CG9883 (Elba2). All three Drosophila proteins lack other recognizable biochemical function protein domains. To date, two crystal structures of the BEN domains of Drosophila proteins complexed with target DNA have been determined: Insv (PDB 4IX7) [39] and Bsg25A (Elba1) (PDB 4 × 0 G) [52]. Analyses revealed the association between the BEN domains and specific DNA binding through extensive nucleotide contacts with their α helices and C-terminal loop, suggesting that the BEN domain functions in a DNA sequence-specific manner. On the other hand, recently, the Drosophila BEN-solo protein Elba2, but not Insv, was shown to rescue the deficiency of histone H1 protein in early larvae [53], supporting the sequence-recognition specificity and sequence-specific function of BEN domains.

The human genome encodes nine BEN domain-containing proteins; NAC1 and NAC2 also contain a BTB domain, whereas the seven other proteins lack other recognizable domains. Human NAC1 and NAC2 are similar (55/64% identity/similarity in overall amino acid sequence), and their BEN domains share 86/92% identity/similarity. Xuan et al. clearly showed that human NAC2 protein binds to non-palindromic TGTCA/GG/CT/AT/AC/TT/CGA/TC sequences through the BEN domain [35]. Immunofluorescence microscopy of human MCF-7 cells with anti-NAC2 polyclonal antibody yields homogeneous uniform staining throughout the nucleus, with the exception of the nucleoli [35]. On the other hand, immunofluorescence analysis of human HeLa cells using a specific monoclonal antibody revealed that endogenous NAC1 is almost exclusively localized with a fine granular pattern in the nucleus, excluding the nucleoli [41]. This fine granular appearance strongly suggests that NAC1 is not uniformly distributed along the chromatin fiber, and our live cell photobleaching analyses revealed that a substantial fraction of NAC1 in the nucleus is associated with or interacts with nuclear proteins or chromatin [42]. These observations prompted us to investigate whether NAC1 directly binds DNA, and if so, which domain of NAC1 is responsible for binding and which consensus DNA-binding sequence human NAC1 binds to. Our bacterial genome carry-over assays revealed that NAC1 directly binds DNA via the BEN domain (Appendix A), similar to NAC2 [35]. In vitro PCR-assisted random oligonucleotide selection experiments showed that monomeric NAC1 preferentially binds ACATGT palindromes containing the core sequence CATG (Figure 1), revealing a DNA-binding property distinct from that of NAC2. Interestingly, the Drosophila BEN-solo proteins Insv and Bsg25A (Elba1) efficiently bind the palindromic CCAATTGG motif in vitro [39,52].

Currently, genome-wide binding data from chromatin immunoprecipitation followed by sequencing (ChIP-seq) experiments provide optimum information for inferring the DNA-binding affinity of transcription factors [54,64]. The NAC1 DNA-binding motif determined from in vitro PCR-assisted random oligonucleotide selection experiments does not match the de novo consensus sequence of the motif identified by DREME and TOMTOM programs in the NAC1 ChIP-seq data of mouse ES cells [18]. This discrepancy is likely due to binding being affected by the specific cell type, condition, developmental stage or tissue, since transcription factors primarily bind DNA in conjunction with one or more other DNA-binding transcription factor and nuclear proteins in cells. Genome-wide binding data from NAC1 ChIP-seq experiments, when coupled with information on the in vitro DNA-binding motif, provide insights into the physical mechanisms of transcriptional modulation of NAC1.

Surprisingly, we observed isotherms for the binding of BEN domain of NAC1 (322-485) with dsDNA oligonucleotide GP1 that show two sequential and non-equivalent binding events (Figure 2b and Appendix A). A molecular interpretation of the two binding events is that the second binding invokes the sequence specific base recognition involving in structural conformational change of protein. We found supportive results for this observation in the NMR structure of BEN domain of NAC1 without target DNA compared with two crystal structures of the BEN domains of Drosophila BEN-sole proteins with target DNA. The superimpositions clearly showed three structural differences (Figure 3D). First, NAC1 has an additional N-terminal α helix, α0 (345-351), outside the canonical BEN domain (374-471). Second, superimposition of the loops connecting α3 and α4 with or without target DNA fit poorly. The loops of Insv and Bsg25A are positioned in the minor grove of the target DNA. Thr300 and Lys312 in Drosophila Insv and Ser298 and Arg310 in Drosophila Bsg25A contact the sugar-phosphate backbone of the target DNA, and Ser304 in Insv and Ser302 in Bsg25A make base-specific contacts (PDB 4IX7 and 4 × 0 G). The loop of NAC1 contains at least three residues (Thr418, Lys430, and Ser422) that can interact with DNA (Figure 3E). Third, the α5 helices of Insv and Bsg25A positioned within the major groove are longer than the α5 helix of the BEN domain of NAC1 in the absence of target DNA. The C-terminal half of α5 helix of Insv harbors Asp351 and Lys352, and Asp349 and Lys352 in Bsg25A: the C-terminal halves of both proteins make base-specific contacts. Furthermore, Lys357 in Insv and Arg355 in Bsg25A form contacts with the sugar-phosphate backbone (PDB 4IX7 and 4 × 0 G). The short α5 helix of NAC1 is followed by Asn466, Arg469, and Arg472 (Figure 3E). The latter two structural differences on the basis of the superimpositions and NMR CSPs (Figure 3C) suggest that α2, α3, and the short α5 helices of NAC1 are involved in the first binding event with the target DNA, and that this first binding facilitates the fixation of the loop connecting α3 and α4 to the target DNA and the formation of a longer α5 helix during the second binding event.

Live cell photobleaching analyses clearly showed that there was no difference in the nuclear mobility and kinetic properties of the BEN domain of NAC1 (251-527) and NAC1 (251-527, L432N) carrying a mutation which abolishes DNA-binding activity (Figure 4). The BEN domain of NAC1 alone could bind the target DNA in vitro (Figure 1, Figure 2 and Figure 3) but was unable to interact with chromatin/other proteins in cells. Our previous study showed that NAC1 in the nucleus is associated with or interacts with nuclear proteins or chromatin [59]. Additional, uncharacterized binding partners may be required for the function of NAC1 in cells. The results also suggest caution in the general and indiscriminate use of the BEN domains of BEN-domain containing proteins in reporter assays in cells.

In summary, NAC1 is a sequence-specific DNA-binding protein that binds the target DNA through the BEN domain. We used molecular and structural biology approaches to clarify the two-step binding model. Furthermore, the results of live cell photobleaching analyses suggest that the BEN domains of BEN-domain containing proteins must be used with caution in reporter assays in cells.

## Figures and Tables

**Figure 1 biomedicines-08-00608-f001:**
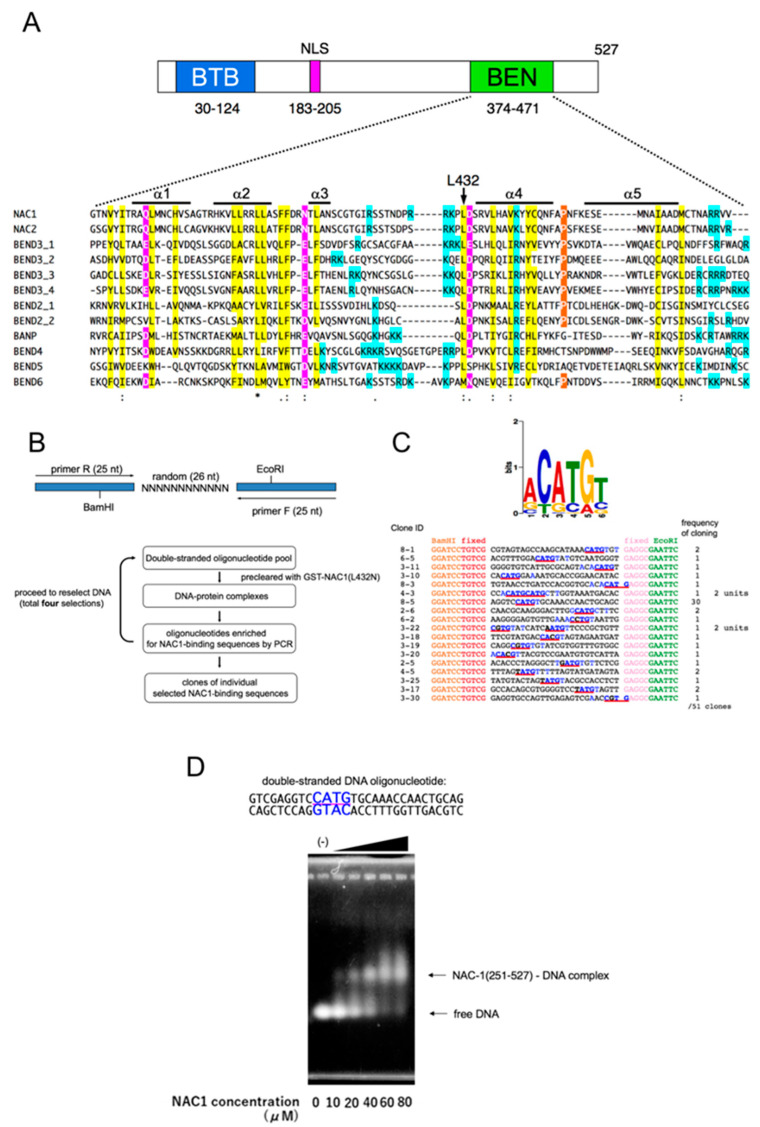
Identification of the consensus DNA-binding sequence of NAC1. (**A**) Schematic representation of human NAC1 protein showing the BTB domain, nuclear localization signal (NLS) and the BEN domain. Clustal Omega alignment of human BEN domain-containing proteins showing highly conserved residues. Color coding reflects the conservation of amino acid types: yellow, blue and magenta for hydrophobic, positively and negatively charged amino acids, respectively. Symbols are as follows: (*), identical residues; (:), highly conserved residues; (.) lower but significant conservation between all members. α-Helical regions are labeled above and are based on Figure 3A. The source and corresponding UniProt accession numbers are indicated. NAC1, Q96RE7; NAC2, Q96BF6; BANP, Q8N9N5; BEND2, Q4V9S2; BEND3, Q5T5 × 7; BEND4, Q6ZU67; BEND5, Q7L4P6; BEND6, Q5SZJ8; BEND7, Q8N7W2. (**B**) Construction of the dsDNA oligonucleotide pool containing a random 26 bp core flanked by 25 nucleotide (nt) primer sequences (upper panel). Lower panel, outline of the selection procedure conducted to determine the DNA-binding motif of NAC1. (**C**) MEME graphical representation of the oligonucleotides selected by NAC1 after four rounds of selection (upper panel). The MEME program parameters were set to use each nucleotide once and to generate a motif with a maximum length of 26 nucleotides. The horizontal axis represents the position of each base. The height of each stack on the vertical axis indicates nucleotide conservation at that position (in bits) and the height of each nucleotide symbol within a stack corresponds to how frequently that nucleotide occurs. Lower panel, sequences of the 18 oligonucleotides used to generate the motif in the upper panel. Only the random portion of the oligonucleotide sequence flanked by the BamHI and EcoRI sites is shown. Sequences corresponding to CATG are underlined in red, and conserved sequences are highlighted with blue characters. The numbers at the left indicate the clone ID. Numbers showing the frequency of cloning are indicated on the right. (**D**) Band shift assay using a biotin-labeled dsDNA oligonucleotide probe corresponding to the #8-5 sequence (Upper panel). Lower panel, the probe (20 μM) was mixed with increasing amounts of bacterially expressed and purified NAC1 without GST (-, 0 μM; 10 μM, 20 μM, 40 μM, 60 μM and 80 μM) and analysed by band shift assay.

**Figure 2 biomedicines-08-00608-f002:**
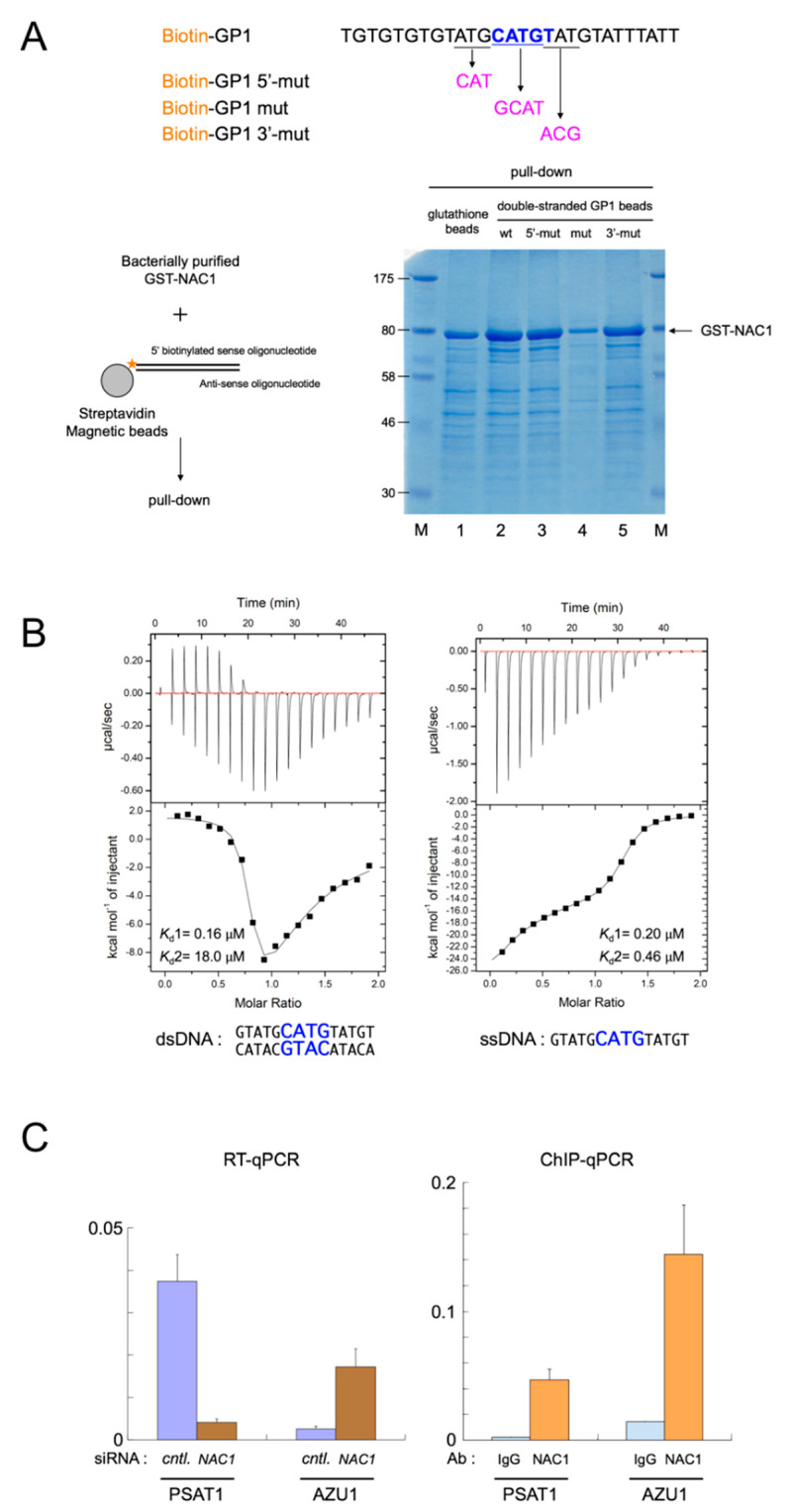
Characterization of the consensus DNA-binding sequence of NAC1. (**A**) Pull-down assay using dsDNA oligonucleotides for wild type and mutant probes against GADD45GIP1 promoter (Upper panel, only sense oligonucleotides are shown). Lower left panel, outline of the pull-down procedure. Lower right panel, 5 μg of bacterially expressed and purified GST-NAC1 was mixed with biotin-labeled dsDNA oligonucleotides for wild type and mutant probes, and pull-down assays were performed. The gel was stained with Coomassie blue to detect NAC1 in the dsDNA-protein complexes pulled down by the oligonucleotides. (**B**) Isothermal titration calorimetry. Upper panels are raw titration data plotted as heat (μcal/s) versus time (min). Each experiment consisted of 28 injections of 10 μL of 50 μM dsDNA (left panel) or of ssDNA (right panel) into a solution of 500 μM NAC-1 (322–485) at 25 °C. The lower panels are integrated heat responses plotted as normalized heat per mole of injectant. Smooth curves represent best fits of the data to the equation as described under “Materials and methods” using software provided by the instrument manufacturer. Data shown is representative of three independent experiments. (**C**) Left panel, RT-qPCR of PSAT1 and AZU1 mRNAs in control (cntl.) and NAC1 knockdown OV207 cells. Right panel, qPCR of ChIP analysis for control IgG (IgG) and NAC1 in OV207 cells.

**Figure 3 biomedicines-08-00608-f003:**
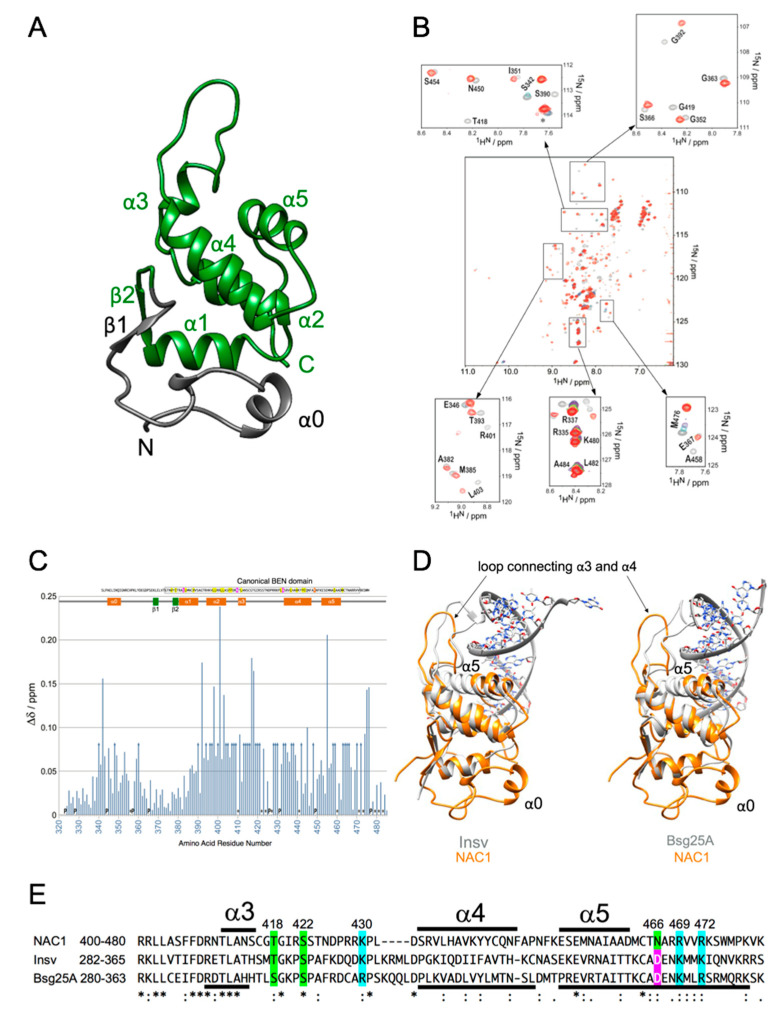
NMR experiments of NAC1. (**A**) Ribbon representation of the lowest-energy solution structure of the BEN domain from human NAC1 (residues 322–485), determined by NMR spectroscopy. The secondary structure elements are shown with arrows (β-strand) and helices (α-helix). The predicted BEN domain is colored green. (**B**) NMR titration experiments of NAC1 with the NAC1 consensus dsDNA oligonucleotide. Overlay of the 1H–15N HSQC spectra of NAC1(322-485) alone (grey) and in the presence of 0.2 (purple), 0.4 (cyan), 0.6 (magenta), 0.8 (green), 1.0 (yellow), 1.2 (pink), and 1.5 (red) equivalents of dsDNA oligonucleotide. The five insets show peaks that disappeared or gradually changed position (from grey to red) in selected spectral regions upon the addition of dsDNA oligonucleotide. (**C**) Total changes in 1HN and 15N chemical shifts (Δδ) for NAC1(322-485) upon the addition of dsDNA oligonucleotide to a ratio of [protein]:[dsDNA] = 1:1.5 are plotted versus residue number. Δδ is given by Δδ = [(ΔδHN)2 + (ΔδN/6.5)2]1/2, where ΔδHN and ΔδN are the chemical shift differences for 1HN and 15N, respectively. Residues that were not assigned (asterisks) or whose 1H-15N resonances disappeared upon the addition of dsDNA oligonucleotide (arrows) are indicated. “P” indicates proline residues. Highly conserved residues of human BEN domain-containing proteins identified by Clustal Omega alignment are highlighted and shown at the top of the plot. See Figure 1A. The canonical BEN domain is boxed. Color coding reflects the conservation of amino acid types: yellow and magenta for hydrophobic and negatively charged amino acids, respectively. (**D**) Structural alignment of the BEN domain of human NAC1 and Drosophila BEN-solo proteins. NAC1 (322-485) is shown as a gold ribbon, and the Insv-BEN domain with dsDNA (PDB ID: 4IX7) (left panel) and Bsg25A (Elba1) with dsDNA (PDB ID: 4 × 0 G) (right panel) are shown in grey. (**E**) Clustal Omega alignment of the BEN domains of human NAC1, Drosophila Insv and Drosophila Bsg25A proteins. Color coding reflects the conservation of amino acid types: green, blue and magenta for hydrophilic, positively and negatively charged amino acids, respectively. Symbols are as follows: (*), identical residues; (:), highly conserved residues; (.) lower but significant conservation between all members. The α-helical regions labeled above are based on Figure 3A and those labeled below are based on PDB 4IX7 and 4 × 0 G. The source and corresponding UniProt accession numbers are indicated. NAC1, Q96RE7; Insv, Q8SYK5; Bsg25A (ELBA1), Q9VR17.

**Figure 4 biomedicines-08-00608-f004:**
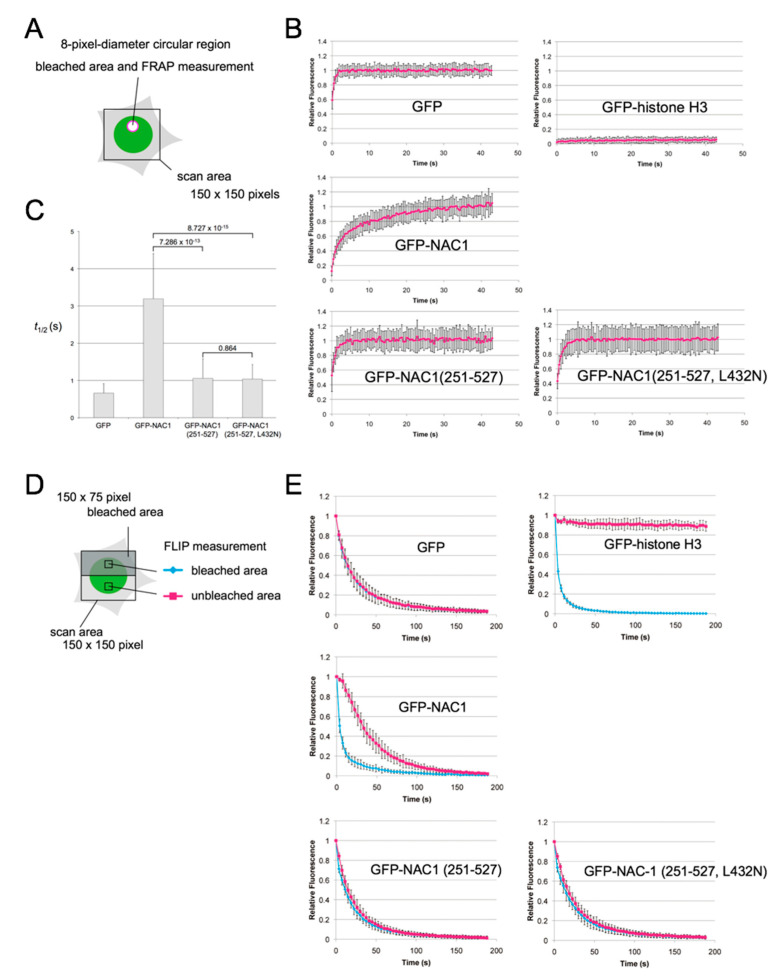
Intranuclear dynamics of the BEN domain of NAC1 in living cells. (**A**) Schematic drawing of FRAP analysis. Quantitative FRAP analysis (FRAP recovery curves) (**B**) and t1/2 analysis (**C**) of stably expressed GFP, GFP-histone H3, GFP-NAC1, GFP-NAC1(251-527) and GFP-NAC1(251-527, L432N) in HeLa cells. Each value is the mean ± SD. Calculated with Welch’s two sample *t* test. (**D**) Schematic drawing of FLIP analysis. (**E**) Quantitative FLIP analysis of stably expressed GFP, GFP-histone H3, GFP-NAC1, GFP-NAC1(251-527) and GFP-NAC1(251-527, L432N) in HeLa cells. The fluorescence loss was monitored in the bleached (blue) and control unbleached (magenta) regions. Each value is the mean ± SD.

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
