# Peer review of "Nucleus Accumbens-Associated Protein 1 Binds DNA Directly through the BEN Domain in a Sequence-Specific Manner"

_biomedicines, 2020, doi:10.3390/biomedicines8120608_

Round 1
Reviewer 1 Report
Urano et al. have discovered NAC1 can bind to DNA directly on the BEN domain with sequence specificity. Authors used PCR to confirm the 6 bp motif ACATGT as a target sequence. The interaction between NAC1 and ACATGT was characterized by various of technologies including gel shift assays, pull-down assays, ITC, chromatin-immunoprecipitation assays. They also applied NMR chemical shifts perturbation (CSP) to investigate the binding structure. Cell culture was used to analyze the binding situation between NAC1 and DNA genome in living cells. Authors are telling a good story about it.
Minor comments:
- For the Figure 1d, I think adding the concentrations just under the gel lane would be better than in the legends.
- Have the authors done any other binding affinity experiments for comparison of ITC. Such as, SPR or Fluorescence binding researches. That will be a helpful support to this manuscript.
Author Response
I wish to express our appreciation to the Reviewer for his or her insightful comments, which have helped us significantly improve the paper.
I would like to answer reviewer’s minor comment.
Minor comments:
For the Figure 1d, I think adding the concentrations just under the gel lane would be better than in the legends.
Have the authors done any other binding affinity experiments for comparison of ITC. Such as, SPR or Fluorescence binding researches. That will be a helpful support to this manuscript.
Answer:
Thank you very much for your suggestion. I added the concentrations just under the gel lane in figure 1d as the reviewer suggested.
We also appreciate your suggestion regarding binding affinity experiments. However, we did not perform any other binding affinity experiments for comparison of ITC.

Reviewer 2 Report
Reviewers comments:
Naomi Nakayama and colleague’s manuscript “Nucleus accumbens-associated protein 1 binds DNA directly through the BEN domain in a sequence-specific manner” study explore the NAC1 protein DNA binding abilities and mapping the domains of interactions. Using conventional approaches such as gel shift assays, pull-down assays, isothermal titration calorimetry (ITC), chromatin-immunoprecipitation assays and NMR chemical shifts perturbation (CSP) authors have characterized the interaction between NAC1 and target DNA.
In summary, using molecular and structural biology approaches authors have shown NAC1 is a sequence-specific DNA-binding protein that directly binds the target DNA through the BEN domain.
Author Response
I wish to express our appreciation to the Reviewer for his or her insightful comments.
As you mentioned in your comment, we have shown NAC1 is a sequence-specific DNA-binding protein that directly binds the target DNA through the BEN domain by using molecular and structural biology approaches. We hope this research will arouse readers’ interest.
Sincerely yours
Naomi Nakayama